# The effect of prioritization over cognitive-motor interference in people with relapsing-remitting multiple sclerosis and healthy controls

**Barbara Postigo-Alonso**[1,2]*, **Alejandro Galvao-Carmona**[1,2], **Cristina Conde-Gavilán**[3,4], **Ana Jover**[3,4], **Silvia Molina**[3,4], **María A. Peña-Toledo**[3,4], **Roberto Valverde-Moyano**[3,4], **Eduardo Agüera**[3,4]

**1** Department of Psychology, Universidad Loyola Andalucía. Seville, Spain, **2** Human Neuroscience Lab (HNL), Universidad Loyola Andalucía, Seville, Spain, **3** Neurology Service, Reina Sofía University Hospital, Cordoba, Spain, **4** Maimonides Institute for Research in Biomedicine of Cordoba, (IMIBIC), Cordoba, Spain

* bpostigo@uloyola.es

**Data Availability Statement:** All relevant data are within the paper and its Supporting Information files.

## Abstract

The cognitive-motor interference (CMI) produced by simultaneous performance of a cognitive and a motor task has been proposed as a marker of real-life impairment of people with Multiple Sclerosis (pwMS), yet there is no consensus on the dual task (DT) procedure. This study aimed to compare DT performance of pwMS and healthy controls (HC) under different instructions and to examine its association with neuropsychological and clinical variables. PwMS (N = 23; relapsing-remitting course) and HC (N = 24) completed the cognitive (Verbal Fluency) and motor (walking) tasks under three conditions: independently or as *single task (ST)*, both tasks simultaneously at best capacity or *double prioritization (DT-DP)*, and only the cognitive task at best capacity while walking at preferred speed or *cognitive prioritization (DT-CP)*. Compared to HC, pwMS walked significantly slower and produced less correct words under all conditions. The distance walked by pwMS and HC significantly differed between conditions (DT-CP< DT-DP< ST). PwMS produced more words during ST respective to DT-DP and DT-CP, with no difference between both DT conditions. HC showed no differences in cognitive performance between conditions. Motor and cognitive dual-task costs (DTC) were similar between groups. Only in pwMS, the cognitive DTC of DT-DP was different from zero. CMI measures correlated with neuropsychological, symptomatic, physiological (cognitive event-related potentials) and clinical variables. These results suggest that cognitive performance while walking is impaired in pwMS, but not in HC. CMI over cognitive performance might be a potential early marker of cognitive decline in pwMS, which may be enhanced by the instruction to prioritize both tasks in DT.

## Introduction

Multiple sclerosis (MS) is a neurodegenerative disease that affects the central nervous system, leading to cognitive and motor deficits among others. Between 40–70% of the people with MS

**Funding:** The authors received no specific funding for this work.

**Competing interests:** The authors have declared that no competing interests exist.

(pwMS) exhibit cognitive impairment detectable with neuropsychological evaluation [1,2] and 50–80% have balance and gait dysfunction [3], which might begin early in the disease course [2,3].

Traditionally, cognitive and motor symptoms have been assessed independently. However, in daily life both processes often coexist (e.g. walk while talking to somebody), so it is time to also include in the standard evaluation a dual task (DT) assessment in which a motor and a cognitive task are performed simultaneously, as it might better reproduce the real-life challenges for pwMS [4].

During the DT assessment, it is common that the performance of either one or both tasks decrements compared to the single task (ST) condition, which represents the so called cognitive-motor interference (CMI). To the extent that the tasks in the DT share the same cognitive -and neural- resources, there would be more competence for these limited resources and, hence, more CMI [5]. Consequently, it is important to select the appropriate type and complexity of the tasks for the DT assessment, especially when evaluating a clinical population such as the pwMS, whose cognitive deficits in the first stages of the disease might be mild and even remain unnoticed in clinical evaluations [6,7]. In line with this, by choosing the proper combination of cognitive and motor tasks, the DT might stand as a "brain stress test" [8] and CMI as an early marker of subclinical cognitive decline in pwMS.

Overall, the evidence to date shows that there is CMI over motor parameters in pwMS, but so there is in healthy controls (HC), and the magnitude of CMI does not differ between these groups [9]. However, it is worth noting that the same reduction in performance or CMI when pwMS already perform significantly worse at ST, could represent a greater impact in their everyday-lives. It has further been suggested that there might be a differential pattern of intragroup changes from ST to DT that should be considered for the assessment of CMI [10].

While a variety of tasks have been used for the DT assessment in pwMS, it has been proposed that the Verbal Fluency task while walking might be sensitive and specific to CMI in pwMS. However, more research is needed in this respect, concerning the instructions for the DT assessment, the effect over the cognitive task and including a matched sample of HC [4,9,10].

Considering this evidence, the present study comprised a DT consisting on the combination of walking and Verbal Fluency task, under two DT conditions respective to the instruction of prioritization: both tasks at best capacity or double priority condition (DT-DP) and performing only cognitive task at best capacity while walking at preferred speed or cognitive priority condition (DT-CP).

The present study aimed to: 1) determine the extent to which DT performance affects motor and cognitive parameters in pwMS compared to HC, 2) explore the effect of different instructions of prioritization in the DT in order to guide clinical decision-making regarding the selection of the DT procedure and 3) examine whether CMI over cognitive and motor parameters is associated with neuropsychological, symptomatic, physiological (cognitive event-related potentials) and clinical variables, as one of the interests is to know whether CMI could be used as marker of functional impairment in pwMS.

It was hypothesized that pwMS would have more CMI than HC and that the instructions given for the DT would lead to a gradient in CMI, being the performance better in ST than in any DT, and in DT-CP better than in DT-DP -i.e. highest CMI-, according to the increased demand and level of competence between both tasks for limited cognitive resources.

## Method

The study was conducted in accordance with the ethical standards laid down in the Declaration of Helsinki (1964) and its later amendments. The experimental protocol was approved by

the ethics committee of the Reina Sofía University Hospital (Cordoba, Spain). The procedures of the study were explained to the participants, who provided written informed consent prior to data recollection.

## Participants

This observational case-control study included a convenience sample of 23 pwMS and 24 HC. PwMS were recruited from the neurology service. HC were recruited by word-of-mouth. Both participants with MS and HC were fluent in Spanish. For the pwMS, the most recent Expanded Disease Severity Scale (EDSS) [11] score, disease duration, number of relapses and time from last relapse parameters were extracted from the health record.

The inclusion criteria for pwMS were: (i) neurologist-confirmed diagnosis of relapsing-remitting MS according to the 2010 revised McDonald diagnostic criteria [12,13]; (ii) relapse-free for at least one month; (iii) EDSS score ≤ 6.5 representing the need of bilateral constant help to walk.

Participants from both groups (pwMS and HC) were excluded from the study if: (i) scored equal or above 29 in the Beck Depression Inventory II (BDI-II), which is the cutoff for severe depression [14,15]; (ii) had neurologic disease (other than MS for the pwMS group), psychiatric disease, visual acuity or field deficits, or any musculoskeletal condition that could interfere with the test procedure.

## Experimental procedure

All participants completed demographic questionnaires, DT assessments, neuropsychological tests and symptomatic scales. Additionally, pwMS underwent an electroencephalography (EEG) recording, which was always performed in the last place. The participants could rest between tasks in the DT, between different tests, and between trials in the EEG recording.

Walking assessments (ST and DT) were performed in a quiet hallway. The rest of the assessment (cognitive ST, neuropsychological, symptomatic, and EEG recording) was performed in a small and quiet room. Complete testing took approximately 2.5 hours.

## Dual-task assessment

The DT assessment consisted on the performance of both cognitive (i.e. Verbal Fluency Test) and motor (i.e. walking) tasks under three conditions: independently at best capacity or *single task (ST)* ("say as many words as possible" or "walk as fast as possible"), both tasks simultaneously at best capacity or *DT with double prioritization (DT-DP)* ("walk as fast possible while reciting as many words as possible"), and performing only the cognitive task at best capacity while walking at preferred speed or *DT with cognitive prioritization (DT-CP)* ("say as many words as possible while walking at your preferred speed. The most important is to say as many words as possible"). The tasks were performed for 60 s under every condition and always in the same order (ST cognitive, ST motor, DT-DP, DT-CP). The selection of these two priority conditions and the order for performance was based on a previous study in older adults without dementia [16].

During the phonological Verbal Fluency Test, participants orally generated as many words as possible starting with a certain letter. Based on the "Neuronorma" study in Spanish population, the selected words were respectively for each trial: "P" (ST), "M" (DT-DP), "R" (DT-CP) [17]. One experimenter walked nearby the side of the participant while writing down the words generated. At the same time, a second experimenter walked 1 m after the participant and videotaped the performance from the back. Both instruments were used to obtain the number of correct words uttered by the participant. These experimenters assured safety of the participant while walking.

Participants completed all walking (ST and DT) trials along a 24 m long path including a 0.6 m radius in both ends until told to stop by one of the experimenters, who set the chronometer for each trial at 60 s. In this moment, the experimenter set a mark on the floor for measuring afterwards the total distance walked by the participant in each trial.

Therefore, the dual-task measures included distance walked and the number of correct words generated as direct DT scores. Dual-task cost (DTC) scores were also calculated for the motor and cognitive measures in the DT assessment according to the widely used equation [18]:

$$DTC = \frac{single\ task - dual\ task}{single\ task} x\ 100$$

## Neuropsychological assessment

The neuropsychological assessment included the administration of the Symbol Digit Modalities Test (SDMT) -in its written version- [19], which is recognized as a measure of cognitive processing speed and has been widely used for the cognitive evaluation of pwMS [20]. The Five Digit Test (FDT) was employed for the assessment of executive function [21]. It consists of four trials of increasing controlled attentional processes: reading numbers, counting, choosing to count under incongruent numeric stimuli and, shifting between reading and counting. The FDT yields a measure of time to complete each trial and two indexes of inhibition and flexibility (the lower the index, the better the cognitive process). The Test de Aprendizaje Verbal España-Complutense (TAVEC), which is the Spanish version of the California Verbal Learning Test -CVLT- [22], was included for the evaluation of episodic verbal memory [23] with three primary measures: Total trials 1–5, which will hereinafter be referred to as Immediate recall, Short-delay free recall and Long-delay free recall.

## Symptomatic assessment

The symptomatic assessment included: the BDI-II, which was also used for selection criteria as previously specified [14,15], the Spanish adaptation of Multiple Sclerosis Quality of Life-54 Instrument (MSQOL-54) [24,25] and the Daily Fatigue Impact Scale (D-FIS) [26], which has been proved as a feasible and valid instrument for measuring MS-related fatigue [27].

## EEG recording

EEG recordings were performed for the assessment of brain activity of pwMS in relation to an auditory selective attentional task. During the recording, participants were sitting on chair, inside a quiet room. The recording was performed with Nicolet™ Viking Quest system (Natus Medical Incorporated, San Carlos, CA, USA) by using 4 bipolar channels, referenced to the contralateral mastoid, ground electrode placed on the forehead, band width 0.1–100 Hz and sampling frequency of 256 Hz. Disk-scalp electrodes were placed according to the 10–10 International System [28]: Cz, Pz, P7 and P8.

The Viking software was used for the semiautomatic detection of latency and amplitude parameters of event-related potentials (ERPs): P3, P1 and N1 components.

During the EEG recording participants performed an auditory oddball task, consisting on three blocks with 200 trials each, and a pause of at least 1 minute between blocks. There were two sets of stimuli: a 1000Hz tone as target or oddball stimulus and a 500Hz tone as standard or frequent stimulus. All tones had an intensity from 72–100 Db, a duration of 50-150ms and were separated with a SOA (stimulus onset asynchrony) of 1 second. In total, each block consisted of 50 target and 150 standard stimuli. The recording was performed with eyes opened.

## Statistical analysis

Data were analyzed using Statistica 10 (StatSoft, Tulsa, OK, USA) and IBM SPSS 25 (IBM Corp., Armonk, NY, USA) softwares. Descriptive statistics were generated for two groups: pwMS and HC. Normal data distribution was evaluated with Shapiro-Wilk normality test. To compare demographic characteristics between groups, T-Tests were used for the quantitative variables and Pearson chi-square for the categorical variables. Single sample T-Tests were used to compare the mean motor and cognitive DTC scores against a zero mean (i.e. no cost). The analysis of motor parameters in direct DT scores was performed with a two-factorial repeated measures ANOVA (three conditions x two groups) and Bonferroni's post-hoc test. Wilcoxon Matched Pairs Test were used for the comparison of direct cognitive scores and DTC scores between conditions in each group (intragroup analysis). For between-groups comparisons, either T-Test or Mann-Whitney U Test were used according to the normality of the distribution and significance was adjusted for multiple comparisons with Bonferroni's correction. Effect sizes were calculated for ANOVA's main and interaction effects with partial eta-squared ($\eta^2$) interpreted as small, moderate, and large, based on values of .01, .06, and .14, respectively; and for mean contrasts with Cohen's $d$, which was equally interpreted based on values of 0.2, 0.5 and 0.8, respectively [29]. Correlation analyses between CMI measures with neuropsychological, symptomatic and clinical variables were performed with Spearman's Rho.

## Results

### Description of the sample

23 pwMS and relapsing-remitting course (18 women, 5 men) with a mean ± SD age of 46.03 ± 8.07 years, MS duration of 8.34 ± 6.41 years, time from last relapse of 2.65 ± 2.04 years and EDSS median 2 (interquartile range: 2) (range 0–5.5) were tested. None of them required assistance for walking. (See Tables 1 and 2).

24 healthy volunteers (16 women, 8 men) with a mean ± SD age of 41.39 ± 11.38 years served as HC group. There were no significant differences in age, gender or educational level between pwMS and HC ($p > 0.05$) as shown in Table 1.

In terms of neuropsychological performance, pwMS performed significantly worse than HC in the SDMT [pwMS: 44 ± 14.3; HC: 57.9 ± 13 ($p < 0.01$)]. In contrast, there were no differences between groups in the FDT indexes of inhibition and flexibility, neither on the mean time under each condition of this test after correction for overall speed per participant [30]. No significant differences were found either in performance on the TAVEC ($p > 0.05$) (see Table 1).

Concerning the clinical tests, pwMS scored significantly higher on depression (BDI-II), fatigue (D-FIS) and lower on quality of life under all subscales of the MSQOL-54 (in all cases $p < 0.01$) (see Table 1).

### Comparison between direct DT scores

In contrast to our hypotheses, no interaction effects were found in motor performance relative to Condition (ST, DT-DP, DT-CP) and Group (pwMS and HC) ($F_{2,90} = 1.20$; $p = 0.305$, $\eta^2 = 0.03$). Main effects were found for Condition ($F_{2,90} = 76.07$; $p < 0.001$, $\eta^2 = 0.63$) and Group ($F_{1,45} = 35.39$; $p < 0.001$, $\eta^2 = 0.44$), with Bonferroni's post-hoc test indicating that pwMS walked significantly slower than HC in ST ($p < 0.001$, $d = 1.59$) and both DT conditions (DT-DP: $p < 0.001$, $d = 1.5$; DT-CP: $p < 0.001$, $d = 1.58$). Similarly, both groups walked significantly faster in ST in comparison with DT-DP (pwMS: $p = 0.028$, $d = 0.46$; HC: $p = 0.007$, $d = 0.58$) and DT-CP conditions (pwMS: $p < 0.001$, $d = 1.19$; HC: $p < 0.001$, $d = 1.77$), and in

**Table 1. Comparison of demographic, neuropsychological and symptomatic features of pwMS and HC (mean ± SD).**

| | PwMS (n = 23) | HC (n = 24) | p- value [c] | PwMS adjusted scores | HC adjusted scores |
|---|---|---|---|---|---|
| Age (years) | 46.03 ± 8.07 | 41.39 ± 11.38 | 0.11 | | |
| Gender (f/m) | 18/5 | 16/8 | 0.37 | | |
| Years of education | 12.78 ± 4.13 [5–19] | 14.75 ± 3.26 [6–20] | 0.08 | | |
| SDMT (n correct) | 44 ± 14.33 | 57.96 ± 13.01 | 0.001* | Sc = 8.4 ± 2.9 | Sc = 10.8 ± 3 |
| FDT- *Reading* [a] | 0.67 ± 0.13 | 0.67 ± 0.11 | 0.87 | Pc = 29.7 ± 24.1 | Pc = 50.1 ± 31.3 |
| FDT- *Counting* [a] | 0.72 ± 0.12 | 0.73 ± 0.07 | 0.74 | Pc = 26.1 ± 24.1 | Pc = 48.7 ± 30.2 |
| FDT- *Choosing* [a] | 1.11 ± 0.16 | 1.13 ± 0.11 | 0.57 | Pc = 36.3 ± 28.6 | Pc = 52.5 ± 29.2 |
| FDT- *Shifting* [a] | 1.5 ± 0.26 | 1.47 ± 0.13 | 0.59 | Pc = 39.9 ± 32 | Pc = 52.4 ± 27.6 |
| FDT- *Inhibition* (seconds) | 15.26 ± 9.16 | 13.55 ± 7.26 | 0.36 | Pc = 50 ± 31.5 | Pc = 53.7 ± 28.5 |
| FDT- *Flexibility* (seconds) | 33.22 ± 24.03 | 23.12 ± 8.59 | 0.19 | Pc = 46.3 ± 35.9 | Pc = 56.5 ± 27.3 |
| TAVEC- *Immediate recall (Trials 1–5)* (n correct words) | 54.48 ± 13.08 | 59.21 ± 9.4 | 0.16 | Z = 0.13 ± 1.22 | Z = 0.5 ± 1.06 |
| TAVEC- *Short-delay free recall* (n correct words) | 11.3 ± 4.13 | 12.75 ± 2.33 | 0.45 | Z = 0 ± 1.45 | Z = 0.33 ± 0.92 |
| TAVEC- *Long-delay free recall* (n correct words) | 11.65 ± 3.93 | 13.17 ± 2.27 | 0.27 | Z = -0.35 ± 1.58 | Z = 0.08 ± 1.14 |
| BDI-II (score 0–63) | 15.7 ± 7.86 | 4.5 ± 3.95 | < 0.001* | | |
| D-FIS [b] (score 0–36) | 15 ± 9.53 | 4.33 ± 4.1 | < 0.001* | | |
| MSQOL-54—*Physical health composite* (score 0–100) | 52.38 ± 23.37 | 85.31 ± 8.82 | < 0.001* | | |
| MSQOL-54—*Mental health composite* (score 0–100) | 60.5 ± 21.61 | 86.28 ± 8.91 | < 0.001* | | |
| MSQOL-54—*Overall quality of life* (score 0–100) | 64.85 ± 14.93 | 83.96 ± 11.28 | < 0.001* | | |

Abbreviations: pwMS, people with multiple sclerosis; HC, healthy controls; SDMT, Symbol Digit Modalities Test; FDT, Five Digit Test; TAVEC, Test de Aprendizaje Verbal España Complutense; BDI-II, Beck Depression Inventory II; D-FIS, Daily Fatigue Impact Scale; MSQOL-54, Multiple Sclerosis Quality of Life-54; Sc, scalar score; Pc: percentile; Z, Standard score.

[a] Direct scores shown correspond to the raw time scores (seconds) corrected for speed per participant as in Faust & Balota (1997). Adjusted scores correspond to the Pc for the raw scores (uncorrected for speed).

[b] D-FIS score is missing from one participant (pwMS n = 22).

[c] P-values correspond to comparisons with T-Tests or Mann-Whitney U Tests of direct scores between pwMS and HC.

* p-value ≤ 0.001

DT-DP relative to DT-CP (pwMS: $p = 0.0005$, $d = 0.68$; HC: $p < 0.001$, $d = 1.09$) (see Fig 1 and Table 3).

After Bonferroni's correction for multiple testing (corrected $p < 0.05/3 = 0.0167$), the number of correct words was significantly lower in both DT conditions in pwMS vs HC (DT-DP: $p = 0.0058$, $d = 0.85$; DT-CP: $p = 0.0062$) and, although marginally, also in ST ($p = 0.0164$, $d = 0.73$). The group of pwMS produced significantly (corrected $p < 0.05/3 = 0.0167$) more words in ST than in both DT conditions (DT-DP: $p = 0.0085$; DT-CP: $p = 0.009$), with no significant differences between DT-DP and DT-CP ($p = 0.821$). In contrast, no significant differences were found in the number of correct words produced by HC in ST versus both DT conditions ($p > 0.05$ in all cases) (see Fig 2 and Table 4).

## Comparison between DTC scores

Single sample t-tests with Bonferroni's correction (corrected $p < 0.05/4 = 0.0125$) revealed that the motor DTC was significantly different from zero for pwMS and HC in both conditions DT-DP and DT-CP respective to ST ($p < 0.001$). However, the cognitive DTC was significantly different from a zero constant only for the DT-DP condition in pwMS (DT-DP: $p = 0.0069$; DT-CP: $p = 0.0128$), but not for HC (DT-DP: $p = 0.21$; DT-CP: $p = 0.19$) (see Table 5).

**Table 2. Clinical and physiological (ERPs) features of pwMS.**

|  | PwMS (n = 23) |
|---|---|
| EDSS | mdn 2 (IQR 2) [0–5.5] |
| Disease duration (years) | 8.34 ± 6.41 [1.25–27.08] |
| Number of relapses | 5.70 ± 3.88 [1–15] |
| Time from last relapse (years) | 2.65 ± 2.04 [0.6–8.24] |
| CzP3—Amplitude | 3.29 ± 2.11 [a] |
| CzP3—Latency | 423.59 ± 57.11 [a] |
| PzP3—Amplitude | 2.86 ± 1.59 [a] |
| PzP3—Latency | 413.55 ± 105.88 [a] |
| P07N1 –Amplitude | -2.86 ± 1.95 [a] |
| P07N1- Latency | 170.68 ± 38.56 [a] |
| P08N1 –Amplitude | -2.97 ± 2.13 [a] |
| P08N1 –Latency | 176.77 ± 37.62 [a] |
| P07P1 –Amplitude | 2.57 ± 1.46 [a] |
| P07P1- Latency | 96.86 ± 15.17 [a] |
| P08P1 –Amplitude | 3.03 ± 1.77 [a] |
| P08P1 –Latency | 95.5 ± 15.7 [a] |

Data are displayed as mean ± SD [range], median (interquartile range) [range], or as otherwise indicated.

Abbreviations: pwMS, people with multiple sclerosis; HC, healthy controls; EDSS, Expanded Disability Status Scale.

[a] ERPs data is missing from one participant (pwMS n = 22).

No significant differences were found between pwMS and HC in motor nor cognitive DTC in any of the DT conditions (DT-DP or DT-CP) ($p > 0.05$ in all cases) (see Table 5).

Matched pairs tests (corrected $p < 0.05/2 = 0.025$) showed significant differences in motor DTC of DT-DP vs DT-CP in both groups (pwMS: $p < 0.0001$ and HC: $p = 0.0001$). In contrast,

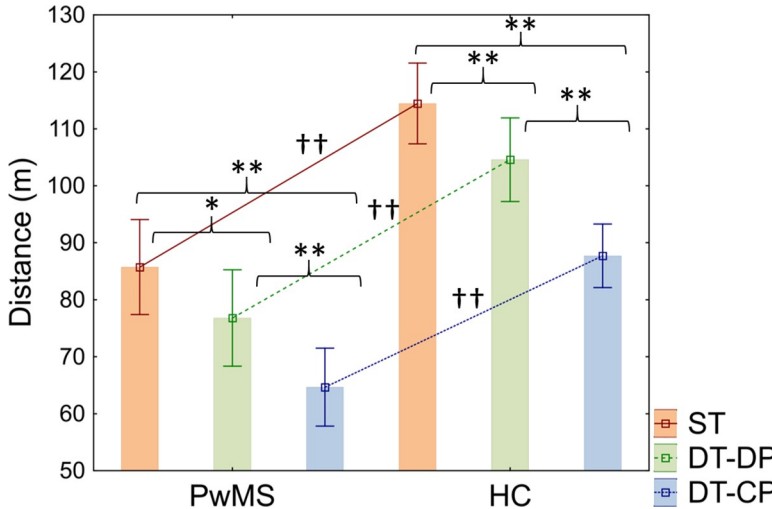

**Fig 1. Cognitive-motor interference over motor performance.** Distance (m) walked (mean ± standard deviation) during single task (ST), dual task with double priority (DT-DP) and dual task with cognitive priority (DT-CP) by people with multiple sclerosis (pwMS) and healthy controls (HC). Significant contrasts are indicated by the black lines over the graph. * $p < 0.05$, ** $p < 0.01$, denoting significant intragroup contrasts -ST vs DT-DP, ST vs DT-CP and DT-DP vs DT-CP- in pwMS and HC, respectively.† $p < 0.05$, †† $p < 0.01$, denoting significant between-group contrasts -pwMS vs HC-.

**Table 3. Intragroup and between-group comparisons of direct scores of motor performance during DT and ST in pwMS and HC (mean ± SD).**

|  | PwMS (n = 23) | HC (n = 24) | Condition | Group | Condition * Group |
|---|---|---|---|---|---|
| ST distance (m) | 85.73 ± 19.3 | 114.45 ± 16.77 | $F = 76.1\ p{<}0.0001^{a,b,c}\ \eta^2 = 0.63$ | $F = 35.4\ p{<}0.0001^{d,e,f}\ \eta^2 = 0.44$ | $F = 1.2\ p = 0.31\ \eta^2 = 0.03$ |
| DT-DP distance (m) | 76.79 ± 19.58 | 104.57 ± 17.38 |  |  |  |
| DT-CP distance (m) | 64.66 ± 15.81 | 87.73 ± 13.24 |  |  |  |

Abbreviations: pwMS, people with multiple sclerosis; HC, healthy controls; ST, single task; DT-DP, dual task with double priority; DT-CP, dual task with cognitive priority.

[a] Post-hoc significant differences between ST and DT-DP in both groups -pwMS and HC- ($p{<}0.05$).

[b] Post-hoc significant differences between ST and DT-CP in both groups -pwMS and HC- ($p{<}0.05$).

[c] Post-hoc significant differences between DT-DP and DT-CP in both groups -pwMS and HC- ($p{<}0.05$).

[d] Post-hoc significant differences in ST between pwMS and HC ($p{<}0.01$).

[e] Post-hoc significant differences in DT-DP between pwMS and HC ($p{<}0.01$).

[f] Post-hoc significant differences in DT-CP between pwMS and HC ($p{<}0.01$).

no differences were found in cognitive DTC of DT-DP vs DT-CP in any of the groups (pwMS: $p{<}0.58$ and HC: $p = 0.85$).

### Relationship between CMI measures & neuropsychology

**Direct DT scores.** The Spearman correlation analysis yielded significant positive correlations between SDMT and all direct DT scores in pwMS ($p \leq 0.01$) except for the distance walked in DT-CP ($p{>}0.05$). In contrast, in HC, the SDMT correlated with correct words in DT-DP (rho = 0.61; $p = 0.001$) and DT-CP (rho = 0.53; $p = 0.008$) and with the distance walked in DT-CP (rho = 0.51; $p = 0.011$). Only in pwMS, a significant negative correlation was observed between FDT-Flexibility and distance in ST (rho = -0.56; $p = 0.005$) and DT-DP (rho = -0.5; $p = 0.015$), correct words in DT-DP (rho = -0.43; $p = 0.041$) and DT-CP (rho = -0.56; $p = 0.006$). The lower the score in FDT-Flexibility indicates higher cognitive flexibility; hence,

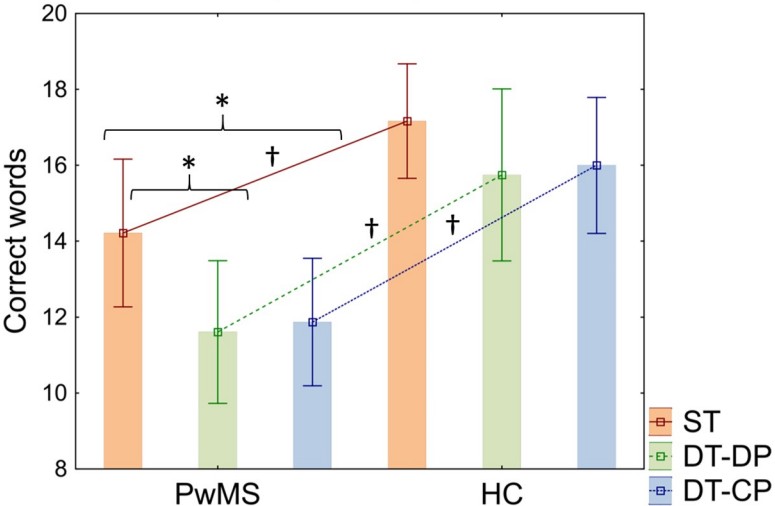

**Fig 2. Cognitive-motor interference over cognitive performance.** Number of correct words uttered (mean ± standard deviation) during single task (ST), dual task with double priority (DT-DP) and dual task with cognitive priority (DT-CP) by people with multiple sclerosis (pwMS) and healthy controls (HC). Significant contrasts are indicated by the black lines over the graph.* Denotes significant intragroup contrasts -ST vs DT-DP, ST vs DT-CP and DT-DP vs DT-CP- in pwMS and HC, respectively. † Denotes significant between-group contrasts -pwMS vs HC-.

**Table 4. Intragroup and between-group comparisons of direct scores of cognitive performance during DT and ST in pwMS and HC (mean ± SD).**

| | ST | DT-DP | DT-CP | ST vs DT-DP | ST vs DT-CP | DT-DP vs DT-CP |
|---|---|---|---|---|---|---|
| **PwMS (n = 23)** | 14.22 ± 4.5 | 11.61 ± 4.35 | 11.87 ± 3.89 | $Z = 2.63$ $p = 0.0085^*$ | $Z = 2.61$ $p = 0.0090^*$ | $Z = 0.23$ $p = 0.8213$ |
| **HC (n = 24)** | 17.17 ± 3.57 | 15.75 ± 5.37 | 16 ± 4.24 | $Z = 1.67$ $p = 0.0944$ | $Z = 1.89$ $p = 0.0582$ | $Z = 0.08$ $p = 0.9353$ |
| **Between-group** | $t = -2.49$ $p = 0.0164^*$ *Cohen's d = 0.73* | $t = -2.9$ $p = 0.0058^*$ *Cohen's d = 0.85* | $U = 147$ $p = 0.0062^*$ | | | |

Abbreviations: pwMS, people with multiple sclerosis; HC, healthy controls; ST, single task; DT-DP, dual task with double priority; DT-CP, dual task with cognitive priority.

\* Denotes significant p-values after Bonferroni's correction for multiple testing.

the more flexibility (lower score), the better motor and cognitive performance in DT (more walked distance and more correct words uttered) in pwMS. No significant correlations were found between FDT-Flexibility and direct DT scores in HC, nor between FDT-inhibition and any DT parameters in any group ($p > 0.05$ in all cases). In addition, there was a significant positive correlation between the TAVEC-Immediate recall and all direct DT scores in pwMS ($p < 0.05$) except for the distance walked in DT-CP ($p > 0.05$), whereas the Long-delay free recall only correlated with the correct words in the three conditions (ST, DT-DP, DT-CP) ($p < 0.05$). In the HC group, only the correct words uttered in ST and DT-CP significantly correlated with TAVEC-Immediate recall, Short-delay free recall and Long-delay free recall ($p < 0.05$).

**DTC scores.** Significant correlations were found only in HC between SDMT and cognitive DTC in DT-DP (rho = -0.44, $p = 0.031$) and between FDT-Inhibition with the cognitive DTC in DT-CP (rho = 0.45; $p = 0.028$). TAVEC- Long-delay free recall showed a significant negative correlation with the cognitive DTC in DT-CP in both groups (pwMS: rho = -0.48, $p = 0.022$; HC: rho = 0.74, $p < 0.001$) (see S1 Table).

## Relationship between CMI measures & symptomatic scales

**Direct DT scores.** Concerning the symptomatic assessment, significant negative correlations were revealed in HC between BDI-II and the distance walked in all conditions (ST: rho = -0.41, $p = 0.047$; DT-DP: rho = -0.52, $p = 0.009$; DT-CP: rho = -0.48, $p = 0.016$), whereas no significant correlations were found between this test and DT parameters in pwMS ($p > 0.05$). On

**Table 5. Comparisons of DTC scores between pwMS and HC (mean ± SD) and contrasts against reference constant (zero value).**

| | PwMS (n = 23) | HC (n = 24) | Between-groups | Single-sample t-test |
|---|---|---|---|---|
| Motor DTC DT-DP | 10.61 ± 9.58 | 8.48 ± 8.77 | $U = 250$; $p = 0.59$ | PwMS: $t = 5.31$; $p < 0.0001^*$<br>HC: $t = 4.74$; $p = 0.0001^*$ |
| Motor DTC DT-CP | 23.37 ± 14.5 | 22.71 ± 10.5 | $U = 537$; $p = 0.76$ | PwMS: $t = 7.73$; $p < 0.0001^*$<br>HC: $t = 10.59$; $p < 0.0001^*$ |
| Cognitive DTC DT-DP | 16.78 ± 26.98 | 7.42 ± 28.47 | $t = 1.16$; $p = 0.25$ *Cohen's d = 0.34* | PwMS: $t = 2.98$; $p = 0.0069^*$<br>HC: $t = 1.28$; $p = 0.21$ |
| Cognitive DTC DT-CP | 14.26 ± 25.23 | 5.75 ± 20.82 | $U = 216.5$; $p = 0.21$ | PwMS: $t = 2.71$; $p = 0.0128$<br>HC: $t = 1.35$; $p = 0.1894$ |

Abbreviations: pwMS, people with multiple sclerosis; HC, healthy controls; ST, single task; DTC DT-DP, dual-task cost in the dual task with double priority; DTC DT-CP, dual-task cost in the dual task with cognitive priority.

\* Denotes significant p-values after Bonferroni's correction for multiple testing.

the other hand, significant negative correlations between D-FIS and the distance walked in ST (rho = -0.54; $p$ = 0.01) and DT-DP (rho = -0.52; $p$ = 0.013) were observed in pwMS, but not in HC ($p > 0.05$). Additionally, significant positive correlations were found between MSQOL-global quality of life and distance walked in all conditions in pwMS (ST: rho = 0.53, $p$ = 0.009; DT-DP: rho = 0.5, $p$ = 0.016; DT-CP: rho = 0.43, $p$ = 0.04), but in HC it only correlated with the distance walked in DT-DP (rho = 0.45; $p$ = 0.028).

**DTC scores.**   Concerning DTC scores, only a significant positive correlation was found between the motor DTC in DT-DP and D-FIS in HC (rho = 0.45; $p$ = 0.026) (see S1 Table).

## Relationship between CMI measures with physiological & clinical variables in pwMS

**Direct DT scores.**   Relative to the ERPs (note that data is missing from one participant (n = 22), P3 amplitude in Cz and Pz significantly correlated with distance in ST (CzP3: rho = 0.49, $p$ = 0.02; PzP3: rho = 0.5, $p$ = 0.018) and with distance in DT-DP (CzP3: rho = 0.45, $p$ = 0.037; PzP3: rho = 0.47, $p$ = 0.028). Latency of P3 in Cz correlated with correct words in DT-CP (rho = 0.43, $p$ = 0.046).

EDSS significantly correlated with distance in ST (rho = -0.66; $p$ = 0.001) and DT-DP (rho = -0.71; $p < 0.001$).

**DTC scores.**   The P3 amplitude in Cz and Pz significantly correlated with the cognitive DTCs in DT-DP (CzP3: rho = -0.43, $p$ = 0.049; PzP3: rho = -0.43, $p$ = 0.044) and in DT-CP (CzP3: rho = -0.49, $p$ = 0.02; PzP3: rho = -0.47, $p$ = 0.028).

No significant correlations were found between the rest of the clinical variables (disease duration, number of relapses, time from last relapse) and any parameters of the DT (direct or DTC) ($p > 0.05$) (see S2 Table).

## Discussion

Our findings confirmed that there is CMI in terms of direct DT scores and DTC over motor performance -i.e. distance walked- in pwMS, and so there is in HC. In contrast, in pwMS there is statistically significant CMI over cognitive performance as well, which is not present in HC. Particularly, pwMS produce less words in DT (DT-DP and DT-CP) than in ST and their cognitive DTC of DT-DP is significantly greater than zero. The instructions of priority had an effect over motor and cognitive performance in this sample.

It should be noted that the sample of present study comprised patients with relapsing-remitting clinical course of MS, in a relatively initial stage of the disease (8.34 ± 6.41 years from diagnosis), free from severe depression, with mild disability [EDSS: median 2 (IQR: 2)] and able to ambulate without assistance.

The presence of CMI over motor parameters in pwMS and HC, together with the fact the motor DTC is comparable between groups is in agreement with current evidence [9,10]. Nonetheless, also in this study, pwMS already had worse motor performance than HC in ST, thus a similar DTC might represent a greater functional impact on the daily life of pwMS.

Despite being overlooked in many cases, the CMI over the cognitive performance in pwMS has been evidenced in other studies as well. As in the present study, pwMS showed significantly reduced performance in DT vs ST conditions -unlike HC- [31,32], and a cognitive DTC significantly greater than zero [33]. However, in our study, no differences were found between cognitive DTC scores of pwMS and HC, in contrast with other studies that identified significant greater cognitive DTC in pwMS than in HC, with no change or even an improvement on a Serial 7s' Subtractions task while walking [32,34]. Of note is that the study by Saleh et al. (2018) comprised a similar pwMS sample, as all patients had relapsing-remitting course of the

disease and similar clinical features. Similarly, in another study, it was found that pwMS performing a digit span task while walking had significant cognitive DTC compared to HC when the number of digits was fixed, but not when it was titrated to each individual's capacity in ST [35]. Thus, it all suggests that CMI over cognitive performance in pwMS is revealed across studies by means of different tasks and measures. The differences in CMI between studies are influenced by the cognitive load coming from the type and complexity of the cognitive tasks concurrent to walking. Moreover, we hypothesize that this is also reason for the differences found between the two DT with different instructions, i.e. DT-DP and DT-CP, as they induce different cognitive loads.

To our knowledge, this is the first study to compare different sets of instructions regarding prioritization of tasks during DT in pwMS. Specifically, in the DT-DP, participants were instructed to perform both tasks at best capacity (walk as fast as possible while reciting as many words as possible), whereas in the DT-CP they were only asked to perform the cognitive task at best capacity while walking at preferred speed. Interestingly, there was a differential pattern of performance between pwMS and HC: while both groups showed significantly reduced motor performance between all conditions according to this gradient (ST > DT-DP > DT-CP), only in pwMS the cognitive performance was significantly reduced from ST to DT-DP and DT-CP respectively, with no significant difference between DT-DP and DT-CP. These results in pwMS are consistent with those obtained in a population of older adults without dementia, who performed the DT consisting on the Alternate Alphabet task while walking under the same conditions of priority -cognitive performance in ST was not assessed- [16]. Additionally, we found that in pwMS and HC, the motor DTC of DT-DP and DT-CP was significant, but only the cognitive DTC of DT-DP was significant in pwMS.

These results suggest that HC successfully prioritized the cognitive task by slowing down in both DT (DT-DP and DT-CP); while pwMS, despite slowing down the same extent as HC, were not able to divert their attention from walking and successfully perform the cognitive task, thus leading to CMI and revealing that their cognitive resources were further exceeded by the DT. In line with this, the CMI over cognitive performance in pwMS was more accentuated in the DT-DP, which is the most cognitively-demanding condition.

Considering this evidence, we would recommend reporting and giving standardized instructions for the DT. Moreover, DT measures are more reliable when participants are given specific instructions of what to prioritize [36]. Specifically, the use of the double-priority instructions would be advised, which might amplify the CMI over motor and cognitive parameters and, therefore, make the DT more sensitive. In support of this, evidence from ST walking have shown that preferred speed is more natural and intuitive [37], but fast walking speed has better metrological properties [38] and has been proposed as more beneficial for the assessment of pwMS because it would rather unveil gait deficits in patients with low EDSS [39].

The neuropsychological, symptomatic scores and clinical features of MS were associated with direct DT and ST measures in a predictable manner, e.g. EDSS and distance in ST and DT-DP. Specifically, it is interesting that cognitive processing speed (SDMT) correlated with all DT scores, except for the distance in DT-CP, in pwMS, suggesting that cognitive processing speed is related to both motor and cognitive processes in ST and DT. However, we replicate the lack of association between SDMT and motor DTC as in previous research [40–42], although it showed a significant association with cognitive DTC in DT-DP in HC. Moreover, greater cognitive flexibility (FDT-Flexibility) was associated exclusively in pwMS with better cognitive performance in both DT-DP and DT-CP -not ST-, and with better motor performance in ST and DT-DP. Therefore, we speculate that the flexibility in allocating cognitive resources to each task might be an important compensatory cognitive process in pwMS for successful performance in DT. Remarkably, no associations were found between clinical

variables of MS such as disease duration like in previous research [43], number of relapses, time from last relapse and any direct or DTC score, suggesting that the symptoms and functional status of the individual are rather related to CMI, independently of these aspects of the disease.

Overall, cognitive DTC scores were associated with various measures of cognition such as processing speed (SDMT) and inhibitory control (FDT- inhibition) in HC, and long-term memory (TAVEC- Long-delay free recall) in pwMS and HC. In addition, cognitive DTC scores in pwMS were associated with physiological measures (ERPs: P3 amplitude in Pz and Cz electrodes). The only significant correlation of motor DTC was with fatigue (D-FIS) in HC during the DT-DP. It should be remarked, because previous studies have not considered the effect over the cognitive task, though it seems to be predominantly associated with the cognitive status of the participants and the motor DTC is not so related to other characteristics of the sample. These results may suggest CMI over cognitive performance as a marker of cognitive status.

In agreement with our results, research found significant associations between CMI and inhibitory measures in HC but not in pwMS, in which it was rather associated with self-perceived difficulty in keeping track of two things at a time [40]. Regarding the physiological results, the P3 amplitude is related with the neural sources required when an attentional task is processed [44], so it can be inferred that the more neural resources are recruited during the attentional task, the better the DT performance.

It is worth noting that different associations have been found with direct DT scores and DTC, which might indicate that they measure different constructs.

The present study is not without limitations. For instance, the sample size is relatively small, and the recruitment was by convenience, which might be a source of bias. Moreover, the mean BDI-II score of pwMS was significantly higher than that of HC, indicative of mild symptomatology of depression at the group level in pwMS. This should be taken into account when considering the results. In addition, the order of the conditions in DT were not counterbalanced. The fact that DT-DP was performed before DT-CP was an informed decision based on previous research [16], but still the ST was always performed prior to DT, so fatigue might have influenced the DT results. Nevertheless, all participants were allowed to rest and sit after each trial or condition. To our knowledge, no previous study has explored the relationship between ERPs and CMI in pwMS. However, the EEG recordings were limited to the pwMS group. Considering that the results were obtained from pwMS with relapsing-remitting course and mild disability, they should not be generalizable to the entire population of pwMS.

Future studies could include other measures of gait performance which might have better captured the effect of CMI, as gait speed or distance have been shown to be sensitive but not specific in pwMS since it also decrements in HC [10].

## Conclusions

The current study examined CMI over cognitive and motor parameters and provided novel data concerning the effect of different instructions of DT prioritization and about its correlates in pwMS and HC. Specifically, it was found that unlike HC, the cognitive performance of pwMS was worse under DT conditions than ST and had a significant cognitive DTC during the DT-DP condition. Furthermore, cognitive DTC scores were associated with neuropsychological and physiological (P3) measures in pwMS. It suggests that CMI over cognitive performance might be a potential early marker of cognitive or functional decline in pwMS, which may be enhanced by the instruction to prioritize both tasks in the DT. Nevertheless, these results should be taken with caution and further research is needed in order to ascertain this question.

## Supporting information

**S1 Dataset.**
(XLSX)

**S1 Table. Correlations between CMI parameters and symptomatic features of pwMS and HC.** Abbreviations: pwMS, people with multiple sclerosis; HC, healthy controls; ST, single task; DT-DP, dual task with double priority; DT-CP, dual task with cognitive priority: DTC, dual-task cost; SDMT, Symbol Digit Modalities Test; FDT, Five Digit Test; TAVEC, Test de Aprendizaje Verbal España Complutense; BDI-II, Beck Depression Inventory II; D-FIS, Daily Fatigue Impact Scale; MSQOL, Multiple Sclerosis Quality of Life-54. Note: Values are Spearman's Rho. D-FIS score is missing from one participant (pwMS n = 22). * p-value < 0.05; ** p-value < 0.001.
(DOCX)

**S2 Table. Correlations between CMI parameters and clinical and physiological features of pwMS.** Abbreviations: pwMS, people with multiple sclerosis; HC, healthy controls; ST, single task; DT-DP, dual task with double priority; DT-CP, dual task with cognitive priority: DTC, dual-task cost. Note: Values are Spearman's Rho. ERPs data is missing from one participant (pwMS n = 22). * p-value < 0.05; ** p-value < 0.001.
(DOCX)

## Author Contributions

**Conceptualization:** Barbara Postigo-Alonso, Alejandro Galvao-Carmona, Eduardo Agüera.

**Data curation:** Barbara Postigo-Alonso, Alejandro Galvao-Carmona, Cristina Conde-Gavilán, Ana Jover, Silvia Molina, María A. Peña-Toledo, Roberto Valverde-Moyano.

**Formal analysis:** Barbara Postigo-Alonso, Alejandro Galvao-Carmona.

**Investigation:** Barbara Postigo-Alonso, Alejandro Galvao-Carmona, Cristina Conde-Gavilán, Ana Jover, Silvia Molina, María A. Peña-Toledo, Roberto Valverde-Moyano.

**Methodology:** Barbara Postigo-Alonso, Alejandro Galvao-Carmona.

**Project administration:** Alejandro Galvao-Carmona, Eduardo Agüera.

**Resources:** Alejandro Galvao-Carmona, Eduardo Agüera.

**Supervision:** Alejandro Galvao-Carmona, Eduardo Agüera.

**Validation:** Barbara Postigo-Alonso, Alejandro Galvao-Carmona, Cristina Conde-Gavilán, Eduardo Agüera.

**Visualization:** Barbara Postigo-Alonso, Alejandro Galvao-Carmona, Eduardo Agüera.

**Writing – original draft:** Barbara Postigo-Alonso.

**Writing – review & editing:** Barbara Postigo-Alonso, Alejandro Galvao-Carmona, Eduardo Agüera.

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
