## [Decision Letter · Decision Letter 0]

14 Nov 2019

PONE-D-19-27815

The effect of prioritization over cognitive-motor interference in people with relapsing-remitting multiple sclerosis and healthy controls

PLOS ONE

Dear Mrs Postigo-Alonso,

Thank you for submitting your manuscript to PLOS ONE. After careful consideration, we feel that it has merit but does not fully meet PLOS ONE’s publication criteria as it currently stands. Therefore, we invite you to submit a revised version of the manuscript that addresses the points raised during the review process.

We would appreciate receiving your revised manuscript by Dec 29 2019 11:59PM. To enhance the reproducibility of your results, we recommend that if applicable you deposit your laboratory protocols in protocols.io, where a protocol can be assigned its own identifier (DOI) such that it can be cited independently in the future. For instructions see: http://journals.plos.org/plosone/s/submission-guidelines#loc-laboratory-protocols

We look forward to receiving your revised manuscript.

Kind regards,

Abiodun E. Akinwuntan, PhD, MPH, MBA

Academic Editor

PLOS ONE

**Reviewers' comments:**

Summary: In this manuscript, researchers investigated the effect of prioritization (single task vs. cognitive prioritization vs. double prioritization) on cognitive and motor outcomes in people with MS compared to the healthy controls. Overall, people with MS had greater cognitive-motor interference compared to healthy controls. In addition, people with MS demonstrated decreased gait speed from a single task to dual-task with double prioritization to dual-task with cognitive prioritization compared to the healthy controls. This study suggests that double prioritization instructions during dual-tasking may improve the cognitive and motor outcomes in people with MS.

In the abstract and introduction, there was no information regarding the rationale of using physiological measure (EEG) to assess brain activity during the auditory oddball test. Please include the rationale and add this to the aims of the study.

Why were the EEG outcomes were not compared with those of healthy controls? In addition, why was the EEG not used during the real-time dual-task conditions? Why was the auditory oddball task selected to measure event-related potentials? Did all participants with people with MS undergo to the EEG assessment? Please clear all these points.

In addition, the physiological assessment was done at the end of the assessment session. Wouldn’t the cognitive and physical fatigue level of the people with MS might affect the event-related potentials? Were there any a priori thoughts to prevent the fatigability of the MS subjects during the EEG assessment?

In the discussion section (page 17, line 382-383), it was stated to the participants to walk as fast as possible while reciting as many words as possible. However, these instructions were not stated in the methods section. If the participants were asked to walk as fast as possible how did the safety was assured? Please add the instructions for walking and the safety precautions in the methods section.

In Table 1, please add the units of the outcomes for the cognitive tests.

In Table 1, it has been shown that there was a significant difference in the BDI-II scores between people with MS and healthy controls. It seems people with MS were in a mild depression. This finding should be considered while interpreting the results. Please incorporate the possibility of the effect of depression on the results in the discussion section.

Please remove the last two sentences of the conclusion which is on page 20 lines 462-463 and incorporate it in the limitations paragraph which is under the discussion section.

Please include a sentence to describe the clinical importance of this study. Should the clinicians instruct double prioritization during activities of living which have a dual-task component in people with MS?

When revising your submission, please upload your figure files to the Preflight Analysis and Conversion Engine (PACE) digital diagnostic tool, https://pacev2.apexcovantage.com/. PACE helps ensure that figures meet PLOS requirements. To use PACE, you must first register as a user. Registration is free. Then, login and navigate to the UPLOAD tab, where you will find detailed instructions on how to use the tool. If you encounter any issues or have any questions when using PACE, please email us at figures@plos.org. Please note that Supporting Information files do not need this step.

---

## [Author Response · Author response to Decision Letter 0]

27 Nov 2019

First of all, we would like to thank the reviewers for the suggestions and recommendations. We really think the comments helped us to improve the paper and make it clearer for readers of the journal. 

Please, note that we will hereinafter refer to pages and lines of the document “Revised Manuscript with Track Changes”, where modifications have been applied.

Comments from Reviewers:

 In the abstract and introduction, there was no information regarding the rationale of using physiological measure (EEG) to assess brain activity during the auditory oddball test. Please include the rationale and add this to the aims of the study.

We wish to thank the reviewer for the suggestion. The rationale for using ERPs with oddball task was to explore if some psychophysiological variables (latency and amplitude of P3, P1 and N1 components) obtained during a cognitive task is related with the performance of a cognitive-motor dual task. In the manuscript, some phrases have been added including the physiological measures:

In the Abstract (page 2, lines 37-38):

CMI measures correlated with “neuropsychological, symptomatic, physiological (cognitive event-related potentials) and clinical variables”.

In the Introduction (page 4, lines 88-89):

3) examine whether CMI over cognitive and motor parameters is associated with “neuropsychological, symptomatic, physiological (cognitive event-related potentials) and clinical variables”, as one of the interests is to know whether CMI could be used as marker of functional impairment in pwMS.” 

Please, note that we have further the specification of “physiological” to the heading in the Results section (page 16, line 341):

“Relationship between CMI measures, with physiological & clinical variables in pwMS”

 Why were the EEG outcomes were not compared with those of healthy controls? In addition, why was the EEG not used during the real-time dual-task conditions? Why was the auditory oddball task selected to measure event-related potentials? Did all participants with people with MS undergo to the EEG assessment? Please clear all these points.

Thank very much for raising these points. The EEG outcomes of pwMS were not compared with that of healthy controls since it did not respond to the research objectives of the study. As can be seen in the manuscript, objective 3 of states: “examine whether CMI over cognitive and motor parameters is associated with neuropsychological, symptomatic, physiological (cognitive event-related potentials) and clinical variables, as one of the interests is to know whether CMI could be used as marker of functional impairment in pwMS.” The physiological evaluation was performed in order to identify possible physiological variables associated with dual-task performance in pwMS.

The oddball paradigm was used during the EEG recording because it is a cognitive paradigm that involves higher-order associative areas of the central nervous system. Furthermore, the oddball paradigm has strong scientific support and the ERP components P1, N1, and P3 evoked during an oddball task are easily and reliably detected. Given that the purpose is to transfer the basic knowledge to the clinic, it is desirable that the simple and widely known paradigms are used for the detection of the ERP components in clinical populations. 

The EEG was not used during real-time dual-task conditions since as mentioned in text (page 8, lines 172-173) the equipment used was the Nicolet™ Viking Quest system (Natus Medical Incorporated, San Carlos, CA, USA), which is not portable. 

Furthermore, the objective was to study the associated cognitive-related variables with DT, rather than the physiological measures underlying DT.

Not all the participants with MS underwent the EEG assessment. As it is noted in the footnotes of table 2 (page 11, line 226) and S2 (page 28, line 620): “ERPs data is missing from one participant (pwMS n=22).” 

A clarifying note is also included in the Results section (page 16, line 343):

Direct DT scores. Relative to the ERPs “(note that data is missing from one participant (n=22)”

 In addition, the physiological assessment was done at the end of the assessment session. Wouldn’t the cognitive and physical fatigue level of the people with MS might affect the event-related potentials? Were there any a priori thoughts to prevent the fatigability of the MS subjects during the EEG assessment?

Thank you for these questions. We agree that cognitive and physical fatigue might have influenced the EEG results. In order to minimize this effect, participants could rest between tasks and between trials during the EEG assessment and, specially, before the EEG assessment they were advised to do so. This has been included in the manuscript as shown below.

We decided to perform it in the last place since it was not the main purpose of the study considering the scare evidence of its relation to dual-task parameters in the literature. So, it was mainly exploratory. Although, the main reason was that the preparation of the subject and the performance of the oddball task is time-consuming and may lead to fatigue, which (in case of doing it in the first place) might as well have influenced performance on the rest of the assessment including the dual task performance. 

A comment was added related to resting in the Experimental procedure subsection of the Method (page 5, lines 119-120) clarifying that: 

All participants completed demographic questionnaires, DT assessments, neuropsychological tests and symptomatic scales. Additionally, pwMS underwent an electroencephalography (EEG) recording, which was always performed in the last place. “The participants could rest between tasks in the DT, between different tests, and between trials in the EEG recording.”

 In the discussion section (page 17, line 382-383), it was stated to the participants to walk as fast as possible while reciting as many words as possible. However, these instructions were not stated in the methods section. If the participants were asked to walk as fast as possible how did the safety was assured? Please add the instructions for walking and the safety precautions in the methods section.

Thank you very much for these suggestions. The instructions are summarized in the Methods (page 6, lines 127-132). We have added a clarification of the instructions in parentheses.

The DT assessment consisted on the performance of both cognitive (i.e. Verbal Fluency Test) and motor (i.e. walking) tasks under three conditions: independently at best capacity or single task (ST) (“say as many words as possible” or “walk as fast as possible”), both tasks simultaneously at best capacity or DT with double prioritization (DT-DP) (“walk as fast possible while reciting as many words as possible”), and performing only the cognitive task at best capacity while walking at preferred speed or DT with cognitive prioritization (DT-CP) (“say as many words as possible while walking at your preferred speed. The most important is to say as many words as possible”). The tasks were performed for 60 s under every condition and always in the same order (ST cognitive, ST motor, DT-DP, DT-CP).

During the walking tasks, the experimenters who walked nearby the side and behind the participant assured safety. 

A sentence about this was included in the Methods section (page 6, line 141): 

One experimenter walked nearby the side of the participant while writing down the words generated. At the same time, a second experimenter walked 1 m after the participant and videotaped the performance from the back. Both instruments were used to obtain the number of correct words uttered by the participant. “These experimenters assured safety of the participant while walking”.

 In Table 1, please add the units of the outcomes for the cognitive tests.

We would like to thank the reviewer for noticing it. The units have been added in Table 1. Please, note that we have also included a clarification for the units of one test in the footnotes of table 1 (page 10, line 216).

 In Table 1, it has been shown that there was a significant difference in the BDI-II scores between people with MS and healthy controls. It seems people with MS were in a mild depression. This finding should be considered while interpreting the results. Please incorporate the possibility of the effect of depression on the results in the discussion section.

We wish to thank the reviewer for this interesting commentary. We agree that this a very important point, so the next phrases were added to the Discussion (page 20, lines 449-452):

“Moreover, the mean BDI-II score of pwMS was significantly higher than that of HC, indicative of mild symptomatology of depression at the group level in pwMS. This should be taken into account when considering the results.”

 Please remove the last two sentences of the conclusion which is on page 20 lines 462-463 and incorporate it in the limitations paragraph which is under the discussion section.

Thank you very much for the recommendation. The sentence has been moved to page 20, lines 457-459.

 Please include a sentence to describe the clinical importance of this study. Should the clinicians instruct double prioritization during activities of living which have a dual-task component in people with MS?

We appreciate this suggestion. The following sentence was added in the Introduction (page 4, lines 86-87):

The present study aimed to: 1) determine the extent to which DT performance affects motor and cognitive parameters in pwMS compared to HC, 2) explore the effect of different instructions of prioritization in the DT “in order to guide clinical decision-making regarding the selection of the DT procedure”.

The clinical recommendations and its importance were also mentioned in:

Discussion (page 18, lines 408-412):

Considering this evidence, we would recommend reporting and giving standardized instructions for the DT. Moreover, DT measures are more reliable when participants are given specific instructions of what to prioritize [36]. Specifically, the use of the double-priority instructions would be advised, which might amplify the CMI over motor and cognitive parameters and, therefore, make the DT more sensitive.

Conclusions (page 21, lines 470-472): 

It suggests that CMI over cognitive performance might be a potential early marker of cognitive or functional decline in pwMS, which may be enhanced by the instruction to prioritize both tasks in the DT.

 We have included an additional correction in the way we designated an equation in the Introduction (page 7, lines 149-150). Previously, it was signaled as:

“Dual-task cost (DTC) scores were also calculated for the motor and cognitive measures in the DT assessment according to the widely used equation [1] [28]:

[1] DTC=(single task-dual task)/(single task) x 100”

However, with this referencing system, the signal “[1]” might be confused with reference number 1, whereas it is only to designate the equation. Therefore, we propose to directly eliminate the “[1]”. It would finally appear like this:

“Dual-task cost (DTC) scores were also calculated for the motor and cognitive measures in the DT assessment according to the widely used equation [28]:

 DTC=(single task-dual task)/(single task) x 100”

---

## [Decision Letter · Decision Letter 1]

6 Dec 2019

The effect of prioritization over cognitive-motor interference in people with relapsing-remitting multiple sclerosis and healthy controls

PONE-D-19-27815R1

Dear Dr. Postigo-Alonso,

We are pleased to inform you that your manuscript has been judged scientifically suitable for publication and will be formally accepted for publication once it complies with all outstanding technical requirements.

Since no modifications are required at this time, you will receive a formal acceptance letter within one week and your manuscript will proceed to our production department and be scheduled for publication.

With kind regards,

Abiodun E. Akinwuntan, PhD, MPH, MBA

Academic Editor

PLOS ONE

---

## [Editor Report · Acceptance letter]

11 Dec 2019

PONE-D-19-27815R1 

The effect of prioritization over cognitive-motor interference in people with relapsing-remitting multiple sclerosis and healthy controls 

Dear Dr. Postigo-Alonso:

I am pleased to inform you that your manuscript has been deemed suitable for publication in PLOS ONE. Congratulations! Your manuscript is now with our production department. 

With kind regards,

on behalf of

Dr. Abiodun E. Akinwuntan 

Academic Editor

PLOS ONE